# Exponential Separation between Two Learning Models and Adversarial Robustness

**Grzegorz Głuch**
EPFL
Lausanne, Switzerland
grzegorz.gluch@epfl.ch

**Ruediger Urbanke**
EPFL
Lausanne, Switzerland
ruediger.urbanke@epfl.ch

## Abstract

We prove an exponential separation for the sample/query complexity between the standard PAC-learning model and a version of the Equivalence-Query-learning model. In the PAC model all samples are provided at the beginning of the learning process. In the Equivalence-Query model the samples are acquired through an interaction between a teacher and a learner, where the teacher provides counterexamples to hypotheses given by the learner. It is intuitive that in an interactive setting fewer samples are needed. We make this formal and prove that in order to achieve an error $\epsilon$ *exponentially* (in $\epsilon$) fewer samples suffice than what the PAC bound requires. It was shown experimentally by Stutz, Hein, and Schiele that adversarial training with on-manifold adversarial examples aids generalization (compared to standard training). If we think of the adversarial examples as counterexamples to the current hypothesis then our result can be thought of as a theoretical confirmation of those findings. We also discuss how our result relates to adversarial robustness. In the standard adversarial model one restricts the adversary by introducing a norm constraint. An alternative was pioneered by Goldwasser et. al. Rather than restricting the adversary the learner is enhanced. We pursue a third path. We require the adversary to return samples according to the Equivalance-Query model and show that this leads to robustness. Even though our model has its limitations it provides a fresh point of view on adversarial robustness.

## 1 Introduction

We analyze the connections between adversarial training, generalization and robustness. Our key technical contribution is a novel result in the field of learning theory, showing an exponential separation for the sample/query complexity of two learning models. This result provides a theoretical confirmation for the experimental observations in Stutz et al. [2019] and points to a fresh research direction in the field of adversarial robustness.

Let us start with a discussion of our main technical contribution. The first model we consider is the PAC-learning introduced in Valiant [1984]. In this model the learner is given i.i.d. samples from an unknown distribution and is tasked with finding a hypothesis with low generalization error. The second model is a version of Equivalence-Query-learning (EQ-learning) introduced in [Angluin, 1988]. In this setting samples are acquired through an interaction between a teacher and a learner. In each round of interaction the learner presents a hypothesis and the teacher replies with a counterexample for this hypothesis. Very close models to the model we study were considered previously in Angluin and Dohrn [2017] and Bshouty and Gavinsky [2002]. In this setting the counterexamples produced by the teacher come from the data distribution constrained to being in the error set of the presented hypothesis. We use ideas similar to those used in boosting techniques [Schapire, 1990] to prove that in the EQ-learning model $O(d \cdot \mathrm{polylog}(1/\epsilon))$ queries suffices to achieve a standard error of $\epsilon$, where $d$ is the VC-dimension of the hypothesis class. This improves on the previously known

35th Conference on Neural Information Processing Systems (NeurIPS 2021), Sydney, Australia.

bounds of $O(d/\epsilon)$. We refer the reader to Section 3 for an in depth review of previous literature on the subject. Standard results for the sample complexity in the PAC model yields an upper bound of $O(d/\epsilon)$. Comparing the two results we see the advertised, exponential (in $\epsilon$), separation.

Let us discuss an "application" of our theorem to the relationship between adversarial robustness and generalization. In [Tsipras et al., 2019, Su et al., 2018] it is argued that there is an inherent trade-off between robustness and accuracy whereas in Gilmer et al. [2018], Rozsa et al. [2016] the authors make the case that robustness improves with accuracy. This seeming contradiction is partially resolved in Stutz et al. [2019] where it is pointed out that it crucially depends on whether adversarial examples are on or off the data manifold. It is worth mentioning that at the beginning of the field of adversarial robustness it was argued that the reason for the existence of adversarial examples is exactly that they are off-manifold [Tanay and Griffin, 2016]. However later on it was demonstrated that adversarial examples do also exist on-manifold [Zhao et al., 2018, Song et al., 2018]. The authors of Stutz et al. [2019] argue that it is important to understand this distinction. They introduce a version of adversarial training [Goodfellow et al., 2014, Madry et al., 2018, Tramèr et al., 2018, Xiao et al., 2019], which they call on-manifold adversarial training, where the adversarial examples are constrained to lie on the data manifold. They then show empirically that using on-manifold adversarial training one achieves better generalization in comparison to standard training (with the same number of samples). Our theorem can be seen as providing a theoretical support for this claim.

We also explore connections of our results to adversarial robustness. Before we describe these connections in detail, let us position this work in a broader context. Most of the literature on adversarial robustness considers adversaries whose perturbations are in some way restricted. A very common form of restriction is to bound the perturbation in the $\ell_0$, $\ell_2$, or $\ell_\infty$ norm [Raghunathan et al., 2018, Wong and Kolter, 2018], but other perturbations (rotations, shifts, etc.) were also considered [Engstrom et al., 2019]. A recent paper [Goldwasser et al., 2020] argues that in order to obtain security for real-world systems we need to consider models beyond these "restricted perturbations." Unfortunately this change comes at a price: e.g., the results in Goldwasser et al. [2020] are based on the assumption that the learner can decide not to give an answer for some inputs (selective learning [Chu, 1965]) and that they see the test set upfront (transductive learning), thus the capabilities of the learner are enhanced. We follow a third path. Instead of restricting the type of perturbations that can be applied by the adversary or enhancing the learner we require the adversary to return counterexamples according to the EQ-model. We then show that the exponential separation theorem leads to robustness in our model in the following way: Th main idea of cryptography is to base security on computational hardness of specific problems. We follows this idea and base the security of our scheme on the hardness of learning. More specifically, our adversarial learning scheme computes a classifier that is robust assuming that the underlying learning task is hard in the low error regime.

## 2 Models of learning and attack

We start by defining the models of learning considered in this paper.

**Notation.** For $i \in \mathbb{N}$ we define $[i] := [1, 2, \ldots, i]$. We use $\log$ to denote the $\log$ to the base 2. For a function $f \to \{-1, +1\}$ and $A \subseteq X$ we define $f \oplus A$ as a function that is equal to $f$ at $X \setminus A$ and flips the prediction for points in $A$. For a distribution $\mathcal{D}$ on $X$ and $A \subseteq X$ we denote by $\mathcal{D}|_A$ the conditional distribution that is equal (up to scaling) to $\mathcal{D}$ on $A$ and $0$ on the complement.

Throughout the paper we consider only the realizable version of learning. That is, we assume that there is a feature space $X$, a distribution $\mathcal{D}$ on $X$, a hypothesis class $\mathcal{H}$ and a function $h \in \mathcal{H}, h : X \to \{-1, +1\}$, that defines the ground truth on $\mathcal{D}$. The *learner* knows $X$ and $\mathcal{H}$ but neither knows $h$ nor $\mathcal{D}$. The goal of the learner is to find an $f : X \to \{-1, +1\}$ that has small risk. The risk is defined as
$$R_{\mathcal{D}, h}(f) := \mathbb{P}_{x \sim \mathcal{D}}[f(x) \neq h(x)].$$
We will consider algorithms that have access to one of the following two oracles:

**Example Query Oracle according to $\mathcal{D}$ (EX$_\mathcal{D}$).** When queried, EX$_\mathcal{D}$ returns $(x, h(x))$, where $x \sim \mathcal{D}$.

**Equivalence Query Oracle according to $\mathcal{D}$ (EQ$_\mathcal{D}$) [Angluin, 1988].** For every $f : X \to \{-1, +1\}$ (not necessarily from $\mathcal{H}$) the result of querying EQ$_\mathcal{D}(f)$ is a counterexample to $f$ dis-

tributed according to $\mathcal{D}$. More formally it is $x \sim \mathcal{D}|_{f \neq h}$. If $R_{\mathcal{D},h}(f) = 0$ then $\text{EQ}_{\mathcal{D}}(f)$ returns "YES" indicating that $f$ is equivalent to $h$. For every $k \in \mathbb{N}$ we write $\text{EQ}_{\mathcal{D}}(f, k)$ to denote an oracle that returns $k$ i.i.d. samples, each generated by $\text{EQ}_{\mathcal{D}}(f)$.

Next we define the two learning models for which we later show the advertised exponential separation. All our results are based on them.

**PAC-learning.** We say that a learning algorithm **L** PAC-learns $\mathcal{H}$ if for every $\epsilon, \delta \in (0, 1)$ there exists $m(\epsilon, \delta) \in \mathbb{N}$ such that for every $h \in \mathcal{H}$ and distribution $\mathcal{D}$ algorithm $\mathbf{L}(\epsilon, \delta)$ asks $m(\epsilon, \delta)$ queries to the Example Query Oracle $\text{EX}_{\mathcal{D}}$ and with probability $1 - \delta$ returns a function $f \in \mathcal{H}$ such that $R_{\mathcal{D},h}(f) \leq \epsilon$.

**EQ-learning** We say that a learning algorithm **L** EQ-learns $\mathcal{H}$ if for every $\epsilon, \delta \in (0, 1)$ there exists $m(\epsilon, \delta) \in \mathbb{N}$ such that for every $h \in \mathcal{H}$ and distribution $\mathcal{D}$ algorithm $\mathbf{L}(\epsilon, \delta)$ asks $m(\epsilon, \delta)$ queries to the Equivalence Query Oracle $\text{EQ}_{\mathcal{D}}$ and with probability $1 - \delta$ returns a function $f : X \to \{-1, +1\}$ such that $R_{\mathcal{D},h}(f) \leq \epsilon$.

## 3  Exponential Separation

We start this section with a short literature review.

It is worth mentioning that there is a slight difference between EQ-learning model and the model from Angluin and Dohrn [2017]. In Angluin and Dohrn [2017] $X$ and $\mathcal{H}$ are finite, the learner has to return the exact hypothesis $h$ that was chosen by the teacher, and the distribution on counterexamples is known, For us, on the other hand, $X$ and $\mathcal{H}$ may be infinite, our goal is to return a function of small risk, and the distribution on counterexamples is not known.

The model from Bshouty and Gavinsky [2002], which is called Probably Almost Exact (PAE) is almost equal to the EQ-learning we consider. The difference is that in the PAE the goal of the learner is to return a hypothesis with negligible error $(1/\omega(\text{poly}(n))$, where $n$ measures measures the "size" of the input. In the EQ-learning on the other hand we want to explore the dependence on $\epsilon$ in detail.

In Angluin and Dohrn [2017] it is shown that for a concept class $|\mathcal{H}| = n$ it is enough to ask $\log(n)$ queries. As we mentioned the setting is different and it's not clear how to adapt this result to the EQ-learning, where we don't need to find the hypothesis exactly (apart from other differences mentioned above). In Bshouty and Gavinsky [2002], where the model considered is closer to ours, it is shown that PAC-learnability $\implies$ PAE-learnability. A direct application of their technique in our setting yields an upper bound on the query complexity, which is no better than $O(d/\epsilon)$. No simple modification of this technique would yield the desired $O(d \cdot \text{polylog}(1/\epsilon))$ query complexity. We significantly extend the ideas from Bshouty and Gavinsky [2002] to obtain our result.

### 3.1  Main result

We are now ready to state the main result of the paper, which is an exponential separation for the sample complexity between PAC-learning and EQ-learning.

**Theorem 1.** *There exists a learning algorithm (Algorithm 1) such that for every $\epsilon \in \left(0, \frac{1}{32}\right), \delta \in (0, 1)$, every hypothesis class $\mathcal{H}$ of VC-dimension $d$, for every distribution $\mathcal{D}$ the algorithm invoked with parameters $\epsilon, \delta, \mathcal{H}$ EQ-learns $\mathcal{H}$ asking*

$$O((d + \log(1/\delta)) \log^9(1/\epsilon)) \text{ queries.}$$

To compare this result to the PAC model we recall that a slightly improved version of the PAC bound in the realizable case [Hanneke, 2016] states that for a hypothesis class $\mathcal{H}$ of VC-dimension $d$ in order to learn (with constant probability) a classifier of risk $\epsilon$ it suffices to use $O\left(\frac{d}{\epsilon}\right)$ samples. This result is tight in a sense that there exist hypothesis classes and distributions for which that many samples are necessary. Theorem 1 guarantees that in the EQ-learning model $O(d \cdot \text{polylog}(1/\epsilon))$ many queries suffice. This is an exponential improvement. Why is this possible?

Imagine that there is a ground truth $g \in \mathcal{H}$ that you try to learn and that you already found $h \in \mathcal{H}$ such that $R_{\mathcal{D},g}(h) \leq \eta$. Then the counterexample oracle $\text{EQ}_{\mathcal{D}}(h, \cdot)$ provides you with samples from a distribution $\mathcal{D}|_{h \neq g}$. Querying the oracle $O(d)$ times you get a sample $S \leftarrow \text{EQ}_{\mathcal{D}}(h, O(d))$. You

can now use the PAC bound: if you find an $h' \in \mathcal{H}$ that is consistent with $S$ then you know that it has an error of at most $1/2$ on $\mathcal{D}|_{h \neq g}$. It is then natural to define

$$\text{Combine}(h, h')(x) = \begin{cases} h(x), & \text{if } x \in h = g \\ h'(x), & \text{if } x \in h \neq g \end{cases}.$$

Note that $\text{Combine}(h, h')$ has an error of at most $\eta/2$. If we repeated this procedure $\log(1/\epsilon)$ times, thus asking $O(d \log(1/\epsilon))$ queries, you would find a classifier with error $\epsilon$.

Unfortunately it is not possible to compute $\text{Combine}(h, h')$. Afterall, if you knew the region of the input space where $h \neq g$ then you could just flip the prediction of $h$ in that region and the resulting classifier would have zero error. But it turns out that the general intuition of decreasing the error by a multiplicative factor after every $O(d)$ queries can indeed be achieved. The key to this result is to find a computable version of $\text{Combine}(h, h', ...)$ that guarantees an exponentially fast decay of the error.

Our algorithm is mainly inspired by boosting techniques, most notably by an approach from Bshouty and Gavinsky [2002]. In this work the authors consider capabilities of polynomially bounded learners. If we assume that the "complexity" of the hypothesis class $\mathcal{H}$ is measured by its VC-dimension then the relevant result from Bshouty and Gavinsky [2002] can be summarized as follows. If we have an algorithm $\mathbf{L}^{\text{PAC}}$ that learns $\mathcal{H}$ to a constant error in time (the authors focus on time but the time is of course an upper-bound for the sample complexity) time$(d)$ then this algorithm can be boosted to an algorithm $\mathbf{L}^{\text{EQ}}$ that learns $\mathcal{H}$ in the EQ-model to an error $\frac{1}{\omega(\text{poly}(d))}$ in time poly(time$(d)$).

Let us apply this boosting technique to our setting. Assume that $\mathbf{L}^{\text{PAC}}$ runs in time $O(d)$ (as this is the number of samples that are required by the standard PAC bound to learn to constant error). Then the boosting algorithm can be used to produce $\mathbf{L}^{\text{EQ}}$ that learns $\mathcal{H}$ to error $\epsilon$. What bound on the run time of $\mathbf{L}^{\text{EQ}}$ do we get? Unfortunately this bound is no better than $O\left(\frac{d}{\epsilon}\right)$. This is exactly what the standard PAC bound provides in the first place. Thus, disappointingly, a direct application of these ideas do not yield a benefit in using the EQ-model versus using the PAC-model.

To get the claimed exponential separation between the two models we develop a boosting-like algorithm that differs significantly in several important aspects and hence also requires a different proof technique. The main idea is to compute $h_1, \ldots, h_{O(\text{polylog}(1/\epsilon))} \in \mathcal{H}$ in a sequential manner and then to define the final hypothesis as a version of a majority vote of these functions.

The simplified, high level, structure of the algorithm is as follows. Repeat the following for $t = O(\text{polylog}(1/\epsilon))$ steps: at step $i$ ask the oracle for $S_i \leftarrow \text{EQ}_{\mathcal{D}}(\text{"Majority"}(h_1, \ldots, h_{i-1}), O(d))$ and then define $h_i := \text{FindConsistent}(S_i, \mathcal{H})$ (FindConsistent returns a function from $\mathcal{H}$ that agrees with all samples from $S_i$). At the end return "Majority"$(h_1, \ldots, h_t)$. To make this approach work we need to ensure that the error sets of $h_1, \ldots, h_t$ are sufficiently independent.

The following points need particular attention. First, note that the $\text{EQ}_{\mathcal{D}}$ provides the algorithm with samples from the error set only. Thus, correctly classified points at one stage will not automatically remain correctly classified at later stages. To make sure this happens, at every step $i$ we include in the training set $S_i$ samples from carefully chosen regions of the feature space that are already classified correctly (this is done in the inner "for" loop of the algorithm). Second, a simple majority vote of the previously constructed classifiers is not sufficient to get the desired result. This is true since a non-negligible region of the feature space might become incorrectly classified with higher and higher confidence by such a majority vote. This is the reason we clip the values of votes to a fixed interval (for details see Definition 1).

We start now with the formal definition of the algorithm.

## 3.2 The Algorithm

In this section we give formal definitions of the concepts used in the algorithm. The proof of correctness of the algorithm is deferred to the appendix. As discussed above we cannot define the final prediction as a simple majority vote. That is why we need to "clip" the votes in a particular way and use a non-standard majority vote-like procedure. Now we define these notions formally.

**Definition 1** (Vote and Majority). *For $\epsilon \in (0, 1)$ we define $B_\epsilon := 2\lceil \log(1/\epsilon) \rceil + 1$ and clip$_\epsilon : \mathbb{Z} \to \mathbb{Z}$ as:*

$$clip_\epsilon(x) := \min(\max(-B_\epsilon, x), B_\epsilon).$$

---
**Algorithm 1** EQlearner
---

**Input:** hypothesis class $\mathcal{H}$ of VC-dimension $d$, target error $\epsilon$, target confidence $\delta$, Equivalence query oracle $EQ_{\mathcal{D}}$.

$\epsilon' := \frac{\epsilon}{10^5 \log^4(1/\epsilon)}$
$B_{\epsilon'} := 2\lceil \log(1/\epsilon') \rceil + 1$
$m := O\left((d + \log(B_{\epsilon'}^4) + \log(1/\delta)) \cdot B_{\epsilon'}^4\right)$
$t := O\left(B_{\epsilon'}^3\right)$
$h_1 \in \mathcal{H}$
**for** $i = 2$ **to** $t$ **do**
    $S_i := EQ_{\mathcal{D}}(\mathrm{Maj}(h_1, \ldots, h_{i-1}), m)$
    **for** $v \in [B_{\epsilon'}] \cap 2\mathbb{Z} + 1$ **do**
        $h' := \mathrm{Maj}(h_1, \ldots, h_{i-1}) \oplus [\mathrm{Vote}(h_1, \ldots, h_{i-1}) \in \{v, -v\}]$
        $S_i^v := EQ_{\mathcal{D}}(h', m)$
    **end for**
    $h_i := \mathrm{FindConsistent}\left(S_i \cup \bigcup_{v \in [B_{\epsilon'}]} S_i^v, \mathcal{H}\right)$
**end for**

**Return** $\mathrm{Maj}(h_1, h_2, \ldots, h_t)$

---

*For a sequence of functions $h_1, \ldots, h_i : X \to \{-1, +1\}$, $\epsilon \in (0, 1)$ and $x \in X$, we define $\mathrm{Vote}(h_1, \ldots, h_i)(x)$ recursively as:*

$$\mathrm{Vote}(h_1, \ldots, h_i)(x) :=$$
$$\mathrm{clip}_\epsilon\left(\mathrm{Vote}(h_1, \ldots, h_{i-1})(x) + 2h_i(x)\right),$$
$$\mathrm{Vote}(h_1)(x) := h_1(x).$$

*Similarly, for a sequence of functions $h_1, \ldots, h_i : X \to \{-1, +1\}$, $\epsilon \in (0, 1)$ and a ground truth function $g$ we define $\mathrm{Vote}_g(h_1, \ldots, h_i)(x)$ recursively as:*

$$\mathrm{Vote}_g(h_1, \ldots, h_i)(x) :=$$
$$\mathrm{clip}_\epsilon\left(\mathrm{Vote}(h_1, \ldots, h_{i-1})(x) + 2 \cdot (-1)^{h_i(x) = g(x)}\right),$$
$$\mathrm{Vote}_g(h_1)(x) := (-1)^{h_1(x) = g(x)}.$$

**Note.** *$\mathrm{Vote}(h_1, \ldots, h_i)(x)$ expresses our current estimate for a particular input (together with a level of confidence), whereas $\mathrm{Vote}_g(h_1, \ldots, h_i)(x)$ denotes the error of this current estimate with respect to the ground truth.*

*Finally, we define:*

$$\mathrm{Maj}(h_1, \ldots, h_i)(x) := \begin{cases} +1, & \text{if } \mathrm{Vote}(h_1, \ldots, h_i)(x) \geq 0, \\ -1, & \text{otherwise.} \end{cases}$$

**Observation 1.** *Observe that for all $i \in \mathbb{N}$, $h_1, \ldots, h_i, g : X \to \{-1, +1\}$ and $x \in X$ we have:*

$$\mathrm{Vote}(h_1, \ldots, h_i)(x) \in 2\mathbb{Z} + 1 \cap [-B_\epsilon, B_\epsilon] \text{ and}$$
$$\mathrm{Vote}(h_1, \ldots, h_i)(x) = \pm\mathrm{Vote}_g(h_1, \ldots, h_i)(x).$$

### 3.3 EQ-learner as a Booster

Guarantees based on the VC-dimension are often not tight and as our result is phrased in these terms one might wonder how much it depends on this specific measure. As mentioned, our algorithm can be understood as a boosting technique and hence the result applies more generally. Next, we explain what we mean by that.

Let $\mathcal{H}$ be a hypothesis class and $\mathcal{D}$ be a distribution. Imagine that we have an algorithm $\mathcal{A}$ that for some distributions learns $\mathcal{H}$. What we mean is that $\mathcal{A}$ is a "PAC-learner" for $\mathcal{H}$ but only for a class of

distributions $\mathfrak{D}$. Then imagine that we use $\mathcal{A}$ as a subroutine in the EQ-learning algorithm. I.e., instead of following the template of $S := EQ_{\mathcal{D}}(f, m), h := FindConsistent(S, \mathcal{H})$ (as in Algorithm 1) we use $\mathcal{A}$ to get an $h$ that has a small error on distribution $EQ_{\mathcal{D}}(f)$. Now assume that all distributions $EQ_{\mathcal{D}}(f)$ for which $\mathcal{A}$ is run belong to $\mathfrak{D}$.

In this case a slight extension of Theorem 1 shows that we can boost $\mathcal{A}$ in the following sense. Assume that for every $\epsilon, \delta \in (0, 1)$, every $\mathcal{D}' \in \mathfrak{D}$ $\mathcal{A}$ learns $\mathcal{H}$ on $\mathcal{D}'$ in $Q_{\mathcal{A}}(\mathcal{H}, \epsilon, \delta)$ number of samples. Then there exists an EQ-learner (this is Algorithm 1, which uses $\mathcal{A}$ as a subroutine) that learns $\mathcal{H}$ up to error $\epsilon$ in number of queries upper-bounded by $Q_{\mathcal{A}}\left(\mathcal{H}, \frac{1}{16}, \frac{\delta}{\text{polylog}(1/\epsilon)}\right) \cdot \text{polylog}(1/\epsilon)$. Now observe that if

$$\frac{Q_{\mathcal{A}}(\mathcal{H}, \epsilon, \delta)}{Q_{\mathcal{A}}\left(\mathcal{H}, \frac{1}{16}, \frac{\delta}{\text{polylog}(1/\epsilon)}\right)} \gg \text{polylog}(1/\epsilon), \tag{1}$$

then the constructed EQ-learner learns $\mathcal{H}$ to error $\epsilon$ with fewer queries than $\mathcal{A}$ does in the PAC-model. This is a different type of separation result. In words it says that in some cases you can boost an algorithm from the PAC-model to the EQ-model such that fewer queries are required. The condition from (1) in words means that the dependence of the runtime of $\mathcal{A}$ on $\epsilon$ grows faster than $\text{polylog}(1/\epsilon)$. This is a reasonable assumption as in the PAC-learning model for every hypothesis one needs $\Omega(1/\epsilon)$ samples just to see a single point from the error set of this hypothesis. This suggests that the dependence of $Q_{\mathcal{A}}(\mathcal{H}, \epsilon, \delta)$ on $\epsilon$ might grow like $\Omega(1/\epsilon)$ (which is exactly what happens in the standard PAC-bound). To summarize, even in the cases when the VC-theory is far from reality one can still hope to get interesting results using our technique.

## 4   On-manifold adversarial training aids generalization

As we mentioned in the introduction, the relationship between adversarial robustness and generalization has been the subject of a robust scientific discussion, no pun intended. We focus here on Stutz et al. [2019] and argue that our result can be interpreted as providing a theoretical confirmation of the important empiricial findings in this paper. As we will see, the setting of Stutz et al. [2019] is close but not identical to ours. But the parallels are strong.

In Stutz et al. [2019] a version of adversarial training called on-manifold adversarial training is proposed. In this scheme at every step one takes a point $x$ sampled from the data distribution $\mathcal{D}$ and generates an adversarial example $x'$, which lies on the data manifold ($x' \in \text{supp}(\mathcal{D})$) and on which the current hypothesis $f$ makes an error. Thus we can associate with this procedure an induced distribution $\mathcal{D}'$ on the error set of $f$, which is a result of the following process: $x \sim \mathcal{D}$, generate adversarial example $x'$.

One of the methods to perform on-manifold adversarial training (as done in Stutz et al. [2019]) is to, first, learn class specific VAE-GANs and then when looking for adversarial examples, search for them in the latent space of the autoencoders.

Note that by construction (if the attacks are successful) $\text{supp}(\mathcal{D}') \subseteq \text{supp}(\mathcal{D}_{f \neq h})$, as adversarial examples are crucially constrained to being on the data manifold and they are points on which $f$ makes errors. But there is also one important distinction. The distributions $\mathcal{D}'$ and $\mathcal{D}_{f \neq h}$ may be different. It is reasonable to assume that they are not too different in many applications and we will proceed under this assumption. With this caveat, the attacker in the on-manifold training procedure can be thought of as being the teacher in our EQ model. Under this assumption, Theorem 1 guarantees that there exists an algorithm that learns faster than in the PAC model.

This is exactly what was experimentally found in Stutz et al. [2019]. When on-manifold adversarial training was used the resulting classifier had lower standard error (generalized better) compared to the standard learning procedure.

**Note.**   We can look at the results from Stutz et al. [2019] from a different perspective. They are an experimental confirmation that Theorem 1 holds, as discussed in Section 3.3, also in situations where VC-theory doesn't describe learnability well. Indeed, for neural networks VC-dimension doesn't determine their learnability as in modern settings the number of parameters in networks is much bigger than the number of samples.

Note that Algorithm 1 is likely not the same as the algorithms that are used for training neural networks. One of the bigger differences is that Algorithm 1 uses a majority vote-like classifier. But in a similar fashion we can think of these experimental results as arguments that Theorem 1 holds for a broader class of learning algorithms.

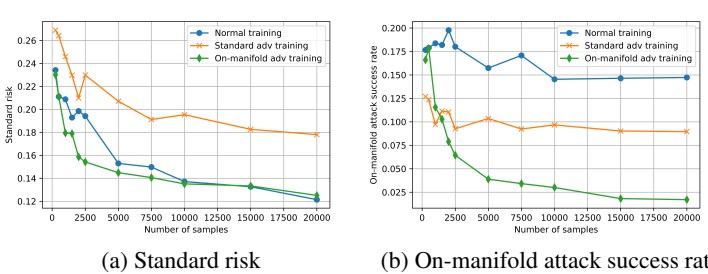

(a) Standard risk  (b) On-manifold attack success rate

Figure 1: Comparison of standard training, adversarial training and on-manifold adversarial training for F-MNIST dataset.

### 4.1 Experiments

For the convenience of the reader we replicate experiments from Stutz et al. [2019], adapting the representation to our needs. Using the code provided by the authors, we reproduced some of their findings on F-MNIST dataset [Xiao et al., 2017]. Results are presented in Figure 1, which reproduces some entries from Figure 5 in Stutz et al. [2019]. In Figure 1 (a) we see that standard risk for on-manifold adversarial training is lower for most settings of the number of samples than for normal training and standard adversarial training. This is consistent with Theorem 1, which states than in a similar model there is an improvement over the PAC-model, which corresponds here to the normal training. We will go back to Figure 1 (b) in Section 5. More details about the experiments is deferred to the appendix.

## 5 Adversarial robustness

Let us now discuss how our results can be seen through the lens of adversarial robustness. We start with a brief review of previous approaches and the lessons we can draw from those.

The standard adversarial learning setup considered in the literature is as follows. There is a classifier $f$ that an adversary $\mathbf{A}$ wants to attack. Having some type of access to $f$, $\mathbf{A}$ generates adversarial examples to $f$. To do so, first a sample $x$ is generated according to $x \sim \mathcal{D}$. Then $\mathbf{A}$ tries to find $x'$ such that $x$ and $x'$ are semantically indistinguishable and $f$ misclassifies $x'$. A formal definition of what "semantically indistinguishable" means is difficult to furnish and it is thus the subject of considerably discussion. The reason for this difficulty is that we expect the machine learning model to learn what "semantically indistinguishable" means from the data and not to decide it by ourselves up front. Many proxies for "semantically indistinguishable" were used in the literature. Usually they come in the form of a restriction on the set of allowed perturbations. As we mentioned in the introduction, some of the most popular restrictions include bounds on the $\ell_0, \ell_2, \ell_\infty$ norm or restrictions to rotations, shifts and many others.

As argued in Tramer and Boneh [2019] defenses often overfit to a particular set of allowed perturbations and the resulting classifiers remain basically undefended against other attacks. This result sparked interest in defenses that make models robust against a wide variety of perturbations, even against unforeseen ones. A recent theoretical result [Goldwasser et al., 2020] proposed a defense that make models robust against all possible perturbations. Unfortunately to achieve this the authors need to allow the learner not to give answers for some inputs and the learner sees the test set before computing the classifier.

As shown in Bubeck et al. [2019] and Tsipras et al. [2019] finding defenses might not be possible even in a simple case of $\ell_2$-bounded perturbations, when the adversary is computationally all powerful. This means that limiting the capabilities of the attacker is very likely to be necessary. Different limitations on the model of attack and the power of the adversary were considered. Early papers limited the type of access that $\mathbf{A}$ has to $f$: instead of full knowledge of $f$ (known as white-box model, see Biggio et al. [2013], Zheng et al. [2019]), a black-box model [Bhambri et al., 2019], partial white-box, where the adversary sees the logits of the output probabilities but doesn't see the internal nodes of a network, oracle access to a gradient of $f$ and others were considered. Unfortunately, even

in the most restrictive model, namely the black-box model, efficient attacks have been shown to exist [Papernot et al., 2017, Chen et al., 2017, Liu et al., 2016, Xiao et al., 2018, Hayes and Danezis, 2017]. This lead the researches to explore models that limit the power of the adversary even further. In Gluch and Urbanke [2020] the authors consider a version of the black-box, where the attacker is limited by the number of evaluations of $f$ it can perform. It is shown that classifiers with high entropy of decision boundaries are hard to attack. In Garg et al. [2020] the authors consider an adversary that is limited computationally. They show that there exist learning problems that can be attacked by an all powerful adversary but are secure against polynomially bounded attackers. These approaches however show security only for some synthetic distributions.

Summarizing this discussion we can formulate two conclusions

1. Introducing new attack models might be necessary for realizing adversarial robustness in the real world.

2. Limiting the power of the adversary is likely inevitable.

## 5.1 Model of attack

As argued in point 1 new attack models might be necessary. We introduce a definition of the adversary as an algorithm that has access to the function it wants to attack and also to the Example Query Oracle for distribution $\mathcal{D}$ and returns points from $X$. More formally:

**Definition 2** (Adversary). *For a feature space $X$ we define an adversary $\mathbf{A}$ as an algorithm[1] (potentially randomized) that for every function $f : X \to \{-1, +1\}$ and $x \in X$ returns $\mathbf{A}(f, x) \in X$. Moreover, for a distribution $\mathcal{D}$, we denote by $\mathbf{A}(f, EX_\mathcal{D})$ a distribution on $X$ that is generated according to a process: sample $x \leftarrow EX_\mathcal{D}$, return $\mathbf{A}(f, x)$. For every $f$ we say that $\mathbf{A}$ is a **restricted adversary** for $f$ if:*

$$\mathbf{A}(f, EX_\mathcal{D}) = EQ_\mathcal{D}.$$

**Note.** We define the adversary as having access to the Example Query Oracle ($EX_\mathcal{D}$). This setting mimics the situation from Section 4 where the adversary was constructing adversarial examples based on samples from the data distribution. Access to $EX_\mathcal{D}$ can be seen as a resource for the adversary that he can but doesn't have to use. Indeed, a priori there is no restriction on the relationship between samples from $EX_\mathcal{D}$ and $\mathbf{A}(f, EX_\mathcal{D})$. Thus we could also assume that $\mathbf{A}$ doesn't have access to $EX_\mathcal{D}$ but posses some approximate knowledge of $\mathcal{D}$ that allows him to satisfy $\mathbf{A}(f, EX_\mathcal{D}) = EQ_\mathcal{D}$.

Finally, let us define the model of attack. It is an adversarial-learning-like scheme where the learning algorithm uses the adversary to learn a more robust model.

**Definition 3** (Adversarial learning game). *For a distribution $\mathcal{D}$ on $X$, a learner $\mathbf{L}$, and an adversary $\mathbf{A}$ we define an adversarial learning game as follows. Learner $\mathbf{L}$ interacts with $\mathbf{A}$ in rounds. In each round $t$ learner $\mathbf{L}$ sends a function $f_t$ to $\mathbf{A}$ and then $\mathbf{A}$ sends a point $x_t \in X$ back to $\mathbf{L}$. At every round $\mathbf{A}$ can query $EX_\mathcal{D}$ once and use the result when generating $x_t$. Decisions made by $\mathbf{L}$ at round $t$ can depend on the messages exchanged before round $t$. For simplicity of the statements we assume that decisions of $\mathbf{A}$ don't depend on the history. At the end of the interaction $\mathbf{L}$ declares a function $f$.*

## 5.2 Interpretation of Theorem 1 for adversarial robustness

We now state a corollary of our main result. The proof is deferred to the appendix:

**Corollary 1.** *For every feature space $X$, for every $\epsilon \in \left(0, \frac{1}{32}\right)$, for every $d \in \mathbb{N}$, for every hypothesis class $\mathcal{H}$ on $X$ of VC-dimension $d$ there exists a learning algorithm $\mathbf{L}$ such that for every distribution $\mathcal{D}$, for every ground truth $h \in \mathcal{H}$, for every adversary $\mathbf{A}$ the following holds. When $\mathbf{L}$ interacts with $\mathbf{A}$ as described in Definition 3 then with probability $2/3$ at least one of the two properties holds:*

- *$\mathbf{L}$, after $O(d \cdot polylog(1/\epsilon))$ rounds of interaction with $\mathbf{A}$, returns a function $f$ such that $R_{\mathcal{D},h}(f) \leq \epsilon$,*

- *there exists $t \in [O(d \cdot polylog(1/\epsilon))]$ such that at the interaction round $t$ a function $f_t$ was presented to $\mathbf{A}$ and $\mathbf{A}(f_t, EX_\mathcal{D}) \neq EQ_\mathcal{D}(f_t)$.*

---

[1] We use *algorithm* here since this seems more natural. But we do not limit the attacker computationally nor are we concerned with questions of computability. Hence, *function* would be equally correct.

*Moreover, throughout the interaction only $O(\mathrm{polylog}(1/\epsilon))$ different functions are presented to $\mathbf{A}$.*

Next we explain how one can understand Corollary 1.

The first point of view is that Corollary 1 explores implications on the existence of restricted adversaries. Interpreted as such, it shows that there exists an adversarial-learning-like scheme where restricted adversaries can be used to learn a hypothesis of error $\epsilon$ exponentially faster than guaranteed by the standard learning theory results.

The second, and arguably the more interesting point of view is the following. If we assume that the adversary we want to defend against is unable to learn a classifier with error $\epsilon$ using $O(d \cdot \mathrm{polylog}(1/\epsilon))$ samples from $\mathcal{D}$ then the adversarial-learning scheme from Corollary 1 can be understood as a defense. It is because throughout the execution of the protocol there was a function $f_t$ that was presented to $\mathbf{A}$ such that $\mathbf{A}(f_t, EX_\mathcal{D}) \neq EQ_\mathcal{D}(f_t)$. Thus the protocol generates a list of $O(\mathrm{polylog}(1/\epsilon))$ many functions such that $\mathbf{A}$ is not a restricted adversary for at least one of them. We believe that this interpretation opens a door for designing defenses that are provably secure against adversaries that are limited by what they can learn. The results of experiments from Figure 1 (b) are consistent with our findings. The probability with which the on-manifold attacks succedes decreases faster than for normal training and standard adversarial training.

**Why it makes sense to assume the learning problem is hard?** As argued in point 2 above, limiting the power of the adversary is most likely unavoidable.

Imagine that the adversary you want to protect your model against is a very good learner. More formally, assume that for a distribution $\mathcal{D}$ and a ground truth $h \in \mathcal{H}$ the adversary can compute a classifier $g$ of very small risk $R_{\mathcal{D},h}(g) \approx 0$. Then when the adversary attacks $f$ it can compute $[f \neq g] \subseteq X$. Note that as the risk of $g$ is close to 0 we know that $\mathbb{P}_{x \sim \mathcal{D}}[f(x) \neq g(x)] \approx R_{\mathcal{D},h}(f)$, that is the region $[f \neq g]$ contains almost all errors of $f$. Even though $\mathbf{A}$ might not know the data distribution $\mathcal{D}$ (we only assumed that $\mathbf{A}$ is able to find a classifier with low error, which doesn't necessarily imply the knowledge of the distribution) it is still able to attack $f$. To do that, for test example $x \sim \mathcal{D}$, $\mathbf{A}$ can find $x' \in [f \neq g]$ that is semantically closest to $x$. Then if the error set of $f$ is semantically close to most of the data distribution then $f$ is indefensible against all powerful learners.

Our result provides an exponential separation between sample/query complexities, as the standard PAC-bound requires $O\left(\frac{d}{\epsilon}\right)$ samples. This resembles the types of separation results, which cryptography is built on and gives hope for following this line of research to find provably secure defenses. Unfortunately this separation doesn't happen for all distributions as the PAC guarantee is only an upper bound. We also know that VC-theory does not necessarily give tight bounds for distributions encountered in practice. We note however that, as discussed in Section 3.3, our result if understood as a boosting technique can be applied to any learning algorithm and thus the implications are not necessarily restricted to the VC-theory.

# 6 Conclusions and Open Problems

We study the interplay between adversarial training, generalization and robustness.

We start by proving an exponential separation between PAC and EQ-learning models. Then we show that our result provides a theoretical justification of previous experimental findings that on-manifold adversarial training helps with generalization. Finally we explore the interpretation of our results for adversarial robustness. We introduce a learning setting where one can use the power of the adversary to evolve a learning scheme to become increasingly robust.

Our result does not yield a provable defense in real-world settings. But it provides a novel point of view on the adversarial robustness problem that hopefully, with further study, will provide such a defense. As discussed the models considered in Section 3 and 4 are different. It would be interesting to prove a version of exponential separation also in the setting from Section 4.

Our result also poses some challenging technical questions. The query complexity upper-bound in the EQ-model we were able to prove is of the form $O\left(d \cdot \log^9(1/\epsilon)\right)$. This is most likely not optimal and what we believe to be the true query complexity is $O(d \cdot \log(1/\epsilon))$. Proving an upper or a lower-bound close to this expression is an interesting theoretical challenge.

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
