# A Proofs

We start by recalling the standard PAC upper-bound for the sample complexity of learning in the realizable case.

**Lemma 1.** *For every hypothesis class $\mathcal{H}$ of VC-dimension $d$ we have that for every $\epsilon, \delta \in (0, 1)$ $\mathcal{H}$ is PAC-learnable using the FindConsistent algorithm with sample complexity:*

$$\frac{d\log(1/\epsilon) + \log(1/\delta)}{\epsilon}.$$

**Lemma 2.** *For every $i \in \mathbb{N}$, for every $h_1, \ldots, h_{i-1} \in \mathcal{H}$, and for every $\delta \in (0, 1)$: if $m = \Omega(d + \log(1/\delta))$ then with probability $1 - \delta$ every function $h \in \mathcal{H}$ that is consistent with $S \sim EQ_{\mathcal{D}}(Maj(h_1, \ldots, h_{i-1}), m)$ satisfies the following:*

$$\mathbb{P}_{x \sim \mathcal{D}}[Maj(h_1, \ldots, h_{i-1})(x) \neq g(x) \wedge h(x) \neq g(x)]$$

$$\leq \frac{1}{16}\mathbb{P}_{x \sim \mathcal{D}}[Maj(h_1, \ldots, h_{i-1})(x) \neq g(x)].$$

In words, the lemma states that if we get $m$ samples and take any function in the hypothesis class that is consistent with those samples, this function will be incorrect at most on a fraction $1/16$ of the error set of our current estimate. Note that the required "independence" of new functions in our boosting-like algorithm is partially satisfied by this statement.

*Proof.* Let $i \in \mathbb{N}$, $h_1, \ldots, h_{i-1} : X \to \{-1, +1\}$. Note that $EQ_{\mathcal{D}}(Maj(h_1, \ldots, h_{i-1}), m)$ generates $m$ i.i.d. samples from the distribution $\mathcal{D}|_{Maj(h_1, \ldots, h_{i-1}) \neq g}$. Then Lemma 1 guarantees that if $m = \Omega(d + \log(1/\delta))$ then with probability $1 - \delta$ every $h \in \mathcal{H}$ that is consistent with $m$ i.i.d. samples from $\mathcal{D}|_{Maj(h_1, \ldots, h_{i-1}) \neq g}$ has error at most $\frac{1}{16}$ on $\mathcal{D}|_{Maj(h_1, \ldots, h_{i-1}) \neq g}$. This is equivalent to the statement of the Lemma. $\square$

**Lemma 3.** *For every $\epsilon' \in (0, 1)$, $i \in [t]$, $h_1, \ldots, h_{i-1} : X \to \{-1, +1\}$, and $v \in [B_{\epsilon'}] \cap 2\mathbb{Z} + 1$ if:*

$$\mathbb{P}_{x \sim \mathcal{D}}[Vote_g(h_1, \ldots, h_{i-1})(x) = -v]$$

$$\geq \frac{1}{B_{\epsilon'}^4}R(Maj(h_1, \ldots, h_{i-1}))$$

*then for $m = \Omega((d + \log(1/\delta))B_{\epsilon'}^4)$ we have that with probability $1 - \delta$ every function $h \in \mathcal{H}$ that is consistent with $EQ_{\mathcal{D}}(h', m)$, where $h' := Maj(h_1, \ldots, h_{i-1}) \oplus [Vote(h_1, \ldots, h_{i-1}) \in \{v, -v\}]$ satisfies the following:*

$$\mathbb{P}_{x \sim \mathcal{D}}[Vote_g(h_1, \ldots, h_{i-1})(x) = -v \wedge h(x) \neq g(x)]$$

$$\leq \frac{1}{16}\mathbb{P}_{x \sim \mathcal{D}}[Vote_g(h_1, \ldots, h_{i-1})(x) = -v]$$

Recall that our classifier is based on a sequence of classifiers. Each of these classifiers casts a vote. Those votes are tallied and possibly clipped. The final classifier looks at the sign of the vote count. We can think of the vote count as the "confidence" we have in the particular decision. Consider all the points in the feature space that have a particular vote count. Assume that this vote count is negative (correct decision) and that this particular vote count has a large probability mass. The lemma then states the following. If we get $m$ further samples and take any function in the hypothesis class that is consistent with those samples, then this function will be incorrect at most on a fraction $1/16$ of the points with this particular vote count.

*Proof.* Let $i \in [t]$, $v \in [B_{\epsilon'}] \cap 2\mathbb{Z} + 1$ and $S \sim EQ_{\mathcal{D}}(h', m)$. We will show that with high probability the following holds:

$$|\{x \in S : Vote_g(h_1, \ldots, h_{i-1})(x) = -v\}| \geq \frac{m}{8B_{\epsilon'}^4}.$$

Let $X_i$ be a Bernoulli random variable that is equal to 1 if and only if the $i$-th sample from $S$ belongs to the region $Vote_g(h_1, \ldots, h_{i-1})(x) = -v$. These random variables are independent and each has success probability $p$, which we claim is at least:

$$\frac{\mathbb{P}_{\mathcal{D}}[Vote_g(h_1, \ldots, h_{i-1})(x) = -v]}{R(Maj(h_1, \ldots, h_{i-1})) + \mathbb{P}_{\mathcal{D}}[Vote_g(h_1, \ldots, h_{i-1})(x) = -v]}.$$

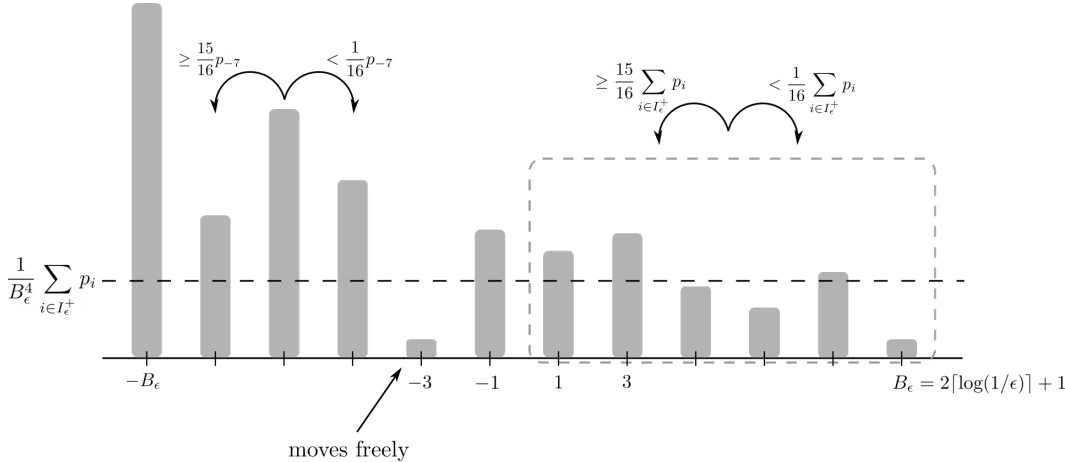

Figure 2: Visualization of the process.

To see that this is true note that for every $x \in X$ such that $\text{Vote}_g(h_1, \ldots, h_{i-1})(x) = -v$ we have by definition of $\text{Vote}_g$ and $\text{Maj}$ that $\text{Maj}(h_1, \ldots, h_{i-1}) = g(x)$. Recall that $h' = \text{Maj}(h_1, \ldots, h_{i-1}) \oplus [\text{Vote}(h_1, \ldots, h_{i-1}) \in \{v, -v\}]$, which means that for every $x$ such that $\text{Vote}_g(h_1, \ldots, h_{i-1})(x) = -v$ we have also that $h'(x) \neq g(x)$, which means that $x$ is misclassified by $h'$. By assumption we have then that

$$p \geq \frac{1}{\left(1 + \frac{1}{B_{\epsilon'}^4}\right) B_{\epsilon'}^4} \geq \frac{1}{2B_{\epsilon'}^4}.$$

We introduce the notation $a \approx_{\beta, \alpha} b$ to denote $a \in [(1 - \beta)b - \alpha, (1 + \beta)b + \alpha]$. By the Chernoff-Hoeffding bound we get that there exists a universal constant $\Gamma$ such that for all $0 < \beta \leq \frac{1}{2}, 0 < \alpha$:

$$\frac{\sum_{i=1}^m X_i}{m} \approx_{\beta, \alpha} p \text{ with probability } 1 - 2e^{-\Gamma k \alpha \beta}.$$

Setting $\beta := \frac{1}{2}, \alpha := \frac{1}{8B_{\epsilon'}^4}$ we get that:

$$\sum_{i=1}^m X_i \geq \frac{m}{8B_{\epsilon'}^4} \text{ with probability } 1 - 2e^{-\frac{\Gamma m}{16 B_{\epsilon'}^4}}.$$

Now observe that conditioned on a sample $x \in S$ being such that $\text{Vote}_g(h_1, \ldots, h_{i-1})(x) = -v$ we know that $x$ is distributed according to $\mathcal{D}|_{\text{Vote}_g(h_1, \ldots, h_{i-1})(x) = -v}$. Thus Lemma 1 guarantees that if $\sum_{i=1}^m X_i \geq O(d + \log(1/\delta))$ then with probability $1 - \delta$ any function consistent with $S$ has error at most $\frac{1}{16}$ on $\mathcal{D}|_{\text{Vote}_g(h_1, \ldots, h_{i-1})(x) = -v}$. So if $m = \Omega((d + \log(1/\delta))B_{\epsilon'}^4)$ then by the union bound over the two failure events we get the result. $\qquad \square$

Next we define an abstract process on odd integers. This process will emulate how a collection of the following probabilities evolves throughout the execution of the algorithm. For iteration $t$ of the algorithm, and a vote value $i \in 2\mathbb{Z} + 1$ we think that $p_i^t$ (which is defined below) is equal to $\mathbb{P}_{x \sim \mathcal{D}}[\text{Vote}_g(h_1, \ldots, h_{i-1})(x) = i]$. The two properties defined in Definition 4 correspond to Lemma 2 and Lemma 3.

**Definition 4** (Process on $2\mathbb{Z}+1$). *For every $\epsilon \in (0, 1)$ we define a process on $I_\epsilon := 2\mathbb{Z}+1 \cap [-B_\epsilon, B_\epsilon]$. For simplicity we introduce the notation $I_\epsilon^+ := I_\epsilon \cap (\mathbb{Z} > 0), I_\epsilon^- := I_\epsilon \cap (\mathbb{Z} < 0)$. For every $i \in I_\epsilon$ and $t \in \mathbb{N}$, there is a value $p_i^t$ associated with a point $i$ at time step $t$. The process starts from an initial configuration $\{p_i^1\}_{i \in I_\epsilon}$, such that $\sum_{i \in I_\epsilon} p_i^1 = 1$. For step $t \in \mathbb{N}$ and for every $i \in I_\epsilon$ the weight $p_i^t$ is split into two parts: a part of $p_i^t$ moves to $i - 2$ and the remaining part moves to $i + 2$. More precisely, this is done in the following manner:*

- *At every step $t$ at least $\frac{15}{16} \sum_{i \in I_\epsilon^+} p_i^t$ of the mass, i.e., at least $\frac{15}{16}$ of the mass on $I_\epsilon^+$, moves down.*

- *At every step $t$ and for every $i \in I_\epsilon^+$, if*

$$p_i^t \geq \frac{1}{B_\epsilon^4} \sum_{i \in I_\epsilon^+} p_i^t$$

*then at most $\frac{1}{16} p_i^t$ of the weight from $p_i^t$ moves to $i+2$.*

*If some mass moved to $-B_\epsilon - 2$ or $B_\epsilon + 2$ then it is moved back to $-B_\epsilon$ and $B_\epsilon$, respectively.*

According to Definition 4, as long as there is any "substantial" mass on a position $i < 0$, at least $15/16$ of this mass has to move two positions down and at most $1/16$ can move two positions up. Moreover $15/16$ of the mass on $i > 0$ has to move down. It is therefore intuitively not surprising that we expect less and less mass to be found on the positive part and the process continues. Lemma 4 makes this intuition quantitative.

Before proving the next Lemma, which gives guarantees on the converges of the process defined in Definition 4, we refer the reader to Figure 2. This figure represent in a visual way the rules of the process. The values $\{p_i\}_{i \in I_\epsilon}$ are arranged on a line, each $p_i$ corresponds to one rectangle. The horizontal dashed line represents the threshold at which the second property from Definition 4 is triggered. The left/right arrows and the values next to them represent how much mass is moved to the left and to the right from a given position.

**Lemma 4.** *Let $\epsilon \in \left(0, \frac{1}{32}\right)$ and consider an initial configuration $\{p_i^1\}_{i \in I_\epsilon}$ such that $\sum_{i \in I_\epsilon} p_i^1 = 1$. Then after $t = O(B_\epsilon^3)$ steps of the process*

$$\sum_{i \in I_\epsilon^+} p_i^t \leq 64 \cdot \epsilon \cdot B_\epsilon^3.$$

To get the final result it is enough to to take the union bound over the failure events of Lemma 2 and 3 and then apply Lemma 4.

**Note.** Optimizing the power on $\log(1/\epsilon)$ in the query upper-bound was not our priority. We focused on simplicity of the algorithm and clarity of the proof. We believe that one can improve the analysis to get a tighter bound. We also think that one would have to come up with a new algorithm to prove that the query complexity of the EQ-model belongs to $o(d \cdot \log^2(1/\epsilon))$.

*Proof.* For $t \in \mathbb{N}$, let $\{p_i^t\}_{i \in I_\epsilon}$ be the configuration resulting from running the process for $t$ steps. We define two metrics that will measure the progress of the process:

$$W_t := \sum_{i \in I_\epsilon^-} 2^i \cdot p_i^t + \sum_{i \in I_\epsilon^+} p_i^t, \text{ and}$$

$$M_t := \frac{\sum_{i \in I_\epsilon^+} i \cdot p_i^t}{\sum_{i \in I_\epsilon^+} p_i^t}.$$

The first metric $W_t$ is a weighted average of these masses, where more weight is put on positions that are "to the left." The second metric $M_t$ is just the expected value of the position of all the weight on the positive part.

Let $t \in \mathbb{N}$. We analyze the evolution from $\{p_i^t\}_{i \in I_\epsilon}$ to $\{p_i^{t+1}\}_{i \in I_\epsilon}$. First note that by definition:

$$\sum_{i \in I_\epsilon^+} p_i^t \leq W_t \leq \sum_{i \in I_\epsilon} p_i^t \leq 1. \tag{2}$$

Observe that $W_t$ is linear in the $\{p_i^t\}$'s. This means that we can analyze the contribution of each weight separately. Let us therefore analyze how the contribution from $p_i^t$ to $W$ changes as $t$ increases to $t + 1$. For every $t \in \mathbb{N}$, any index $i \in I_\epsilon$ belongs to one of the following types:

**Type 1.** $i \in I_\epsilon^- \setminus \{-B_\epsilon\}, p_i^t \geq \frac{1}{B_\epsilon^4} \sum_{j \in I_\epsilon^+} p_j^t$: By definition the contribution of $p_i^t$ to $W_t$ is equal to $2^i \cdot p_i^t$. At step $t + 1$, $p$ of the mass moves to $i - 2$ and $p_i^t - p$ moves to $i + 2$. By the rules of the process (Definition 4) $p \geq \frac{15}{16} p_i^t$. Hence the contribution of this mass to $W_{t+1}$ is at most

$$2^{i-2} \cdot p + \min(2^{i+2}, 1) \cdot (p_i^t - p)$$
$$\leq 2^{i-2} \cdot \frac{15}{16} p_i^t + 2^{i+2} \cdot \frac{1}{16} p_i^t$$
$$\leq 2^{i-2} \cdot p_i^t \cdot \left(\frac{15}{16} + 1\right)$$
$$\leq \frac{1}{2} \cdot 2^i \cdot p_i^t. \tag{3}$$

This means that the **contribution to $W$ decreases by a multiplicative factor of at least** $2$.

**Type 2.** $i \in I_\epsilon^- \setminus \{-B_\epsilon\}, p_i^t < \frac{1}{B_\epsilon^4} \sum_{j \in I_\epsilon^+} p_j^t$: As in the previous case the contribution of $p_i^t$ to $W_t$ is equal to $2^i \cdot p_i^t$. By the rules of the process $0 \leq p \leq p_i^t$ of the mass moves to $i - 2$ and $p_i^t - p$ moves to $i + 2$. Thus the contribution of this mass to $W_{t+1}$ is at most:

$$2^{i-2} \cdot p + \min\left(2^{i+2}, 1\right) \cdot (p_i^t - p) \leq p_i^t. \tag{4}$$

This means that the **contribution to $W$ increases additively by at most** $p_i^t$.

**Type 3.** $i = -B_\epsilon$: The contribution of $p_i^t$ to $W_t$ is equal to $2^{-B_\epsilon} \cdot p_i^t$. By the rules of the process $p \leq p_i^t$ of the mass of $p_i^t$ moves to $i + 2 = -B_\epsilon + 2$. Thus the contribution of this mass to $W_{t+1}$ is at most:

$$2^{-B_\epsilon} \cdot (p_{-B_\epsilon}^t - p) + 2^{-B_\epsilon+2} \cdot p$$
$$\leq 2^{-B_\epsilon+2} \cdot p_{-B_\epsilon}^t$$
$$\leq 2\epsilon^2$$
$$\leq 2\epsilon, \tag{5}$$

where the second to last inequality follows from the fact that $p_{-B_\epsilon}^t \leq 1$ and that $B_\epsilon = 2\lceil \log(1/\epsilon) \rceil + 1$. This means that the **contribution to $W$ increases additively by at most** $2\epsilon$.

**Type 4.** $i \in I_\epsilon^+$: The contribution of $p_i^t$ to $W_t$ is equal to $p_i^t$. By the rules of the process $p \leq p_i^t$ of the mass moves to $i - 2$ and $p_i^t - p$ of the mass moves to $i + 2$ (or stays at $i$ if $i = B_\epsilon$). Thus the contribution of this mass to $W_{t+1}$ is at most:

$$\min\left(2^{i-2}, 1\right) \cdot p + 1 \cdot (p_i^t - p) \leq p_i^t. \tag{6}$$

This means that the **contribution to $W$ decreases**.

Observe that the contribution to $W$ can increase only for Type 2 and 3. Moreover, note that the total amount of mass that can be in Type 2 is at most $B_\epsilon \cdot \frac{1}{B_\epsilon^4} \sum_{j \in I_\epsilon^+} p_j^t \leq \frac{1}{B_\epsilon^3} \sum_{j \in I_\epsilon^+} p_j^t$. Thus combining the observations for the 4 Types we get that:

$$W_{t+1} \leq W_t + 2\epsilon + \frac{1}{B_\epsilon^3} \sum_{j \in I_\epsilon^+} p_j^t.$$

Combined with the left-most inequality in (2), we get

$$W_{t+1} \leq \left(1 + \frac{1}{B_\epsilon^3}\right) W_t + 2\epsilon. \tag{7}$$

Now we analyze the evolution of $W$ from time $t$ to time $t + 1$ depending on how much mass moves from $-1$ to $+1$ and vice versa. Let $0 \leq \delta_{+,-} \leq p_1^t, 0 \leq \delta_{-,+} \leq p_{-1}^t$ be the amount of mass that is moved from $+1$ to $-1$ and from $-1$ to $+1$, respectively. We consider the following cases:

**Case 1.** $\delta_{-,+} \geq \frac{1}{2B_\epsilon} \cdot \sum_{j \in I_\epsilon^+} p_j^t$: We will show that if $W_t \geq 16\epsilon B_\epsilon$, then $W_{t+1} \leq \left(1 - \frac{1}{9B_\epsilon}\right) W_t$.

First, observe that if $p_{-1}^t < \frac{1}{B_\epsilon^4} \sum_{i \in I_\epsilon^+} p_i^t$ then $p_{-1}^t < \frac{1}{2B_\epsilon} \cdot \sum_{j \in I_\epsilon^+} p_j^t \leq \delta_{-,+}$. This means that there is not enough mass on $-1$ to satisfy the assumption of this case. Thus we know that $p_{-1}^t \geq \frac{1}{B_\epsilon^4} \sum_{i \in I_\epsilon^+} p_i^t$. By the rules of the process only $\frac{1}{16}$ of $p_{-1}^t$ can potentially move to $+1$. This implies that $p_{-1}^t \geq 16\delta_{-,+} \geq \frac{16}{2B_\epsilon} \cdot \sum_{j \in I_\epsilon^+} p_j^t$. Using the properties of Type 1 we know that the contribution of $p_{-1}^t$ to $W$ goes down by at least a factor 2.

Let us now look at the evolution of the contribution of $\sum_{i \in I_\epsilon^+} p_i^t$. We know from Type 4 that for this type the contribution does not increase.

Since we know that $p_{-1}^t \geq \frac{8}{B_\epsilon} \cdot \sum_{j \in I_\epsilon^+} p_j^t$, the total contribution of all the weight on $\{-1\} \cup I_\epsilon^+$ goes down by a factor of at least

$$\frac{\frac{2}{B_\epsilon} + 1}{\frac{4}{B_\epsilon} + 1} \leq 1 - \frac{1}{B_\epsilon},$$

where we used that $\epsilon < \frac{1}{32}$.

It remains to include the contributions of the mass at $I_\epsilon^- \setminus \{-1\}$. We know from the properties of Type 1, 2, and 3 that for those the contribution decreases by a factor 2 with the exception of very "small" masses and the mass at the left-hand side boundary. Since $\frac{1}{2} \leq 1 - \frac{1}{B_\epsilon}$ we get:

$$W_{t+1} \leq \left(1 - \frac{1}{B_\epsilon}\right) \cdot W_t + \frac{1}{B_\epsilon^3} \sum_{j \in I_\epsilon^+} p_j^t + 2\epsilon$$

$$\leq \left(1 - \frac{1}{B_\epsilon} + \frac{1}{B_\epsilon^3}\right) \cdot W_t + 2\epsilon, \tag{8}$$

where in the last inequality we used the left-most inequality from (2).

To get the claimed bound, observe that if $W_t \geq 16\epsilon B_\epsilon$, then by (8) we get that

$$W_{t+1} \leq \left(1 - \frac{1}{9B_\epsilon}\right) W_t. \tag{9}$$

**Case 2.** $\delta_{+,-} \geq \frac{1}{2B_\epsilon} \cdot \sum_{j \in I_\epsilon^+} p_j^t$: We will show that if $W_t \geq 16\epsilon B_\epsilon$, then $W_{t+1} \leq \left(1 - \frac{1}{9B_\epsilon}\right) W_t$.

First, observe that the contribution of $p_1^t$ to $W$ decreases by at least $\frac{1}{4B_\epsilon} \cdot \sum_{j \in I_\epsilon^+} p_j^t$. This is true since at least $\frac{1}{2B_\epsilon} \cdot \sum_{j \in I_\epsilon^+} p_j^t$ of the mass was moved from position $+1$ which is weighted by 1 to position $-1$ which is weighted by $\frac{1}{2}$.

Let us now look at the evolution of the contribution of $I_\epsilon^+ \setminus \{+1\}$. We know from Type 4 that for this type the contribution does not increase.

Since we know that the contribution of $p_1^t$ decreased by at least $\frac{1}{4B_\epsilon} \cdot \sum_{j \in I_\epsilon^+} p_j^t$ the total contribution of all the weight in $I_\epsilon^+$ goes down by a factor of at least

$$1 - \frac{1}{4B_\epsilon}.$$

It remains to include the contributions of the mass at $I_\epsilon^-$. We know from the properties of Type 1, 2, and 3 that for those the contribution decreases by a factor 2 with the exception of very "small" masses and the mass at the left-hand side boundary. Since $\frac{1}{2} \leq 1 - \frac{1}{4B_\epsilon}$ we get:

$$W_{t+1} \leq \left(1 - \frac{1}{4B_\epsilon}\right) \cdot W_t + \frac{1}{B_\epsilon^3} \sum_{j \in I_\epsilon^+} p_j^t + 2\epsilon$$

$$\leq \left(1 - \frac{1}{4B_\epsilon} + \frac{1}{B_\epsilon^3}\right) \cdot W_t + 2\epsilon \tag{10}$$

where in the last inequality we used the left-most inequality from (2).

To get the claimed bound, observe that if $W_t \geq 16\epsilon B_\epsilon$ then by (10) we get that

$$W_{t+1} \leq \left(1 - \frac{1}{9B_\epsilon}\right) W_t. \tag{11}$$

**Case 3.** $\delta_{-,+}, \delta_{+,-} < \frac{1}{2B_\epsilon} \cdot \sum_{j \in I_\epsilon^+} p_j^t$: We will show that $M_{t+1} \leq M_t - 1$.

For simplicity we introduce notation $\mu_t := M_t \cdot \sum_{j \in I_\epsilon^+} p_j^t$. First let's analyze what happens when $\delta_{-,+} = 0$. By the rules of the process at least $\frac{15}{16} \sum_{j \in I_\epsilon^+} j \cdot p_j^t$ of the mass on $I_\epsilon^+$ moves down. This and the assumption that $\delta_{+,-} < \frac{1}{2B_\epsilon} \cdot \sum_{j \in I_\epsilon^+} p_j^t$ gives the following two bounds:

$$\mu_{t+1} \leq \mu_t + \left(-2 \cdot \frac{15}{16} + 2 \cdot \frac{1}{16}\right) \cdot \sum_{j \in I_\epsilon^+} p_j^t + \frac{1}{2B_\epsilon} \cdot \sum_{j \in I_\epsilon^+} p_j^t$$

$$\leq \mu_t + \left(\frac{1}{2B_\epsilon} - \frac{7}{4}\right) \cdot \sum_{j \in I_\epsilon^+} p_j^t$$

and

$$\sum_{j \in I_\epsilon^+} p_j^{t+1} \geq \sum_{j \in I_\epsilon^+} p_j^t - \frac{1}{2B_\epsilon} \cdot \sum_{j \in I_\epsilon^+} p_j^t$$

$$= \left(1 - \frac{1}{2B_\epsilon}\right) \sum_{j \in I_\epsilon^+} p_j^t$$

Combining the two bounds we get:

$$M_{t+1} = \frac{\mu_{t+1}}{\sum_{j \in I_\epsilon^+} p_j^{t+1}}$$

$$\leq \frac{\mu_t + \left(\frac{1}{2B_\epsilon} - \frac{7}{4}\right) \cdot \sum_{j \in I_\epsilon^+} p_j^t}{\left(1 - \frac{1}{2B_\epsilon}\right) \sum_{j \in I_\epsilon^+} p_j^t}$$

$$= \frac{2B_\epsilon}{2B_\epsilon - 1} \left(\frac{\mu_t}{\sum_{j \in I_\epsilon^+} p_j^t} - \frac{7}{4}\right) + \frac{1}{2B_\epsilon - 1}$$

$$= \frac{2B_\epsilon}{2B_\epsilon - 1} \left(M_t - \frac{7}{4}\right) + \frac{1}{2B_\epsilon - 1}$$

$$\leq M_t - 1 \tag{12}$$

where in the last equality we used the definition of $\mu_t$. In the last inequality we used the fact that $M_t \leq B_\epsilon$. Note that if $\delta_{-,+} \neq 0$ then $M_{t+1}$ can only decrease as $M_{t+1} \in [1, B_\epsilon]$ and the mass that comes from $-1$ to $+1$ arrives at $1$. Thus by (12) we get that in Case 3:

$$M_{t+1} \leq M_t - 1. \tag{13}$$

**Merging Cases 1, 2 and 3.** First, observe that Cases 1, 2 and 3 cover all potential values of $\delta_{-,+}$ and $\delta_{+,-}$. Then note that by (7) if $W_t \geq 10\epsilon B_\epsilon^3$ then:

$$W_{t+1} \leq \left(1 + \frac{12}{10B_\epsilon^3}\right) W_t \tag{14}$$

Combining (9), (11), (13) and (14) we get that if $W_t \geq \max(10\epsilon B_\epsilon^3, 16\epsilon B_\epsilon) = 10\epsilon B_\epsilon^3$ then either $W_t$ decreases by a multiplicative factor $\left(1 - \frac{1}{9B_\epsilon}\right)$ or $M_t$ decreases by an additive $1$ and $W_t$ increases by at most a multiplicative factor of $\left(1 + \frac{12}{10B_\epsilon^3}\right)$. If on the other hand $W_t < 10\epsilon B_\epsilon^3$ then we have by the definition of $W$ that $W_{t+1} \leq 4W_t$, as each amount of mass can increase it's contribution by at most a multiplicative factor of $4$.

As $M_t \in [1, B_\epsilon]$ it means that in every consecutive $B_\epsilon$ steps rule (9)/(11) is triggered at least once. Thus we have that:

$$W_{t+B_\epsilon} \leq \max\left( \left(1 + \frac{12}{10B_\epsilon^3}\right)^{B_\epsilon} \left(1 - \frac{1}{9B_\epsilon}\right) W_t, \right.$$

$$\left. 40\epsilon B_\epsilon^3 \left(1 + \frac{12}{10B_\epsilon^3}\right)^{B_\epsilon} \right)$$

$$\leq \max\left( e^{\frac{12}{10B_\epsilon^2}} \cdot e^{-\frac{1}{9B_\epsilon}} \cdot W_t, 64\epsilon B_\epsilon^3 \right)$$

$$\leq \max\left( e^{-\frac{1}{81B_\epsilon}} \cdot W_t, 64\epsilon B_\epsilon^3 \right). \tag{15}$$

where in the first inequality we used that either for all $t \in [t, t + B_\epsilon]$ we have that $W_t \geq 10\epsilon B_\epsilon^3$ or there exists $t' \in [t, t + B_\epsilon]$ such that $W_{t'} < 10\epsilon B_\epsilon^3$. The two terms govern the first and the second case respectively. If $W_{t'} < 10\epsilon B_\epsilon^3$ then $W$ grows by a multiplicative factor of at most 4 in every step until it reaches at most $40\epsilon B_\epsilon^3$ and then it grows at most by a multiplicative factor of $1 + \frac{12}{10B_\epsilon^3}$ per step. In the last inequality we used that $\epsilon < \frac{1}{32}$ and that $B_\epsilon = 2\lceil \log(1/\epsilon) \rceil + 1$.

By the right-most inequality of (2) we get that $W_0 \leq 1$, which by using (15) implies that after $t = O(B_\epsilon^3)$ steps $W_t \leq 64\epsilon B_\epsilon^3$, which by the left-most inequality of (2) gives:

$$\sum_{i \in I_\epsilon^+} p_i^t \leq 64\epsilon B_\epsilon^3.$$

$\square$

Now we are ready to prove the main theorem of the paper.

**Theorem 1.** *There exists a learning algorithm (Algorithm 1) such that for every $\epsilon \in \left(0, \frac{1}{32}\right), \delta \in (0, 1)$, every hypothesis class $\mathcal{H}$ of VC-dimension $d$, for every distribution $\mathcal{D}$ the algorithm invoked with parameters $\epsilon, \delta, \mathcal{H}$ EQ-learns $\mathcal{H}$ asking*

$$O((d + \log(1/\delta)) \log^9(1/\epsilon)) \text{ queries.}$$

*Proof.* Observe that the number of queries $q$ asked by the algorithm is upper bounded by:

$$q \leq t \cdot (B_{\epsilon'} + 1) \cdot m$$
$$\leq O\left(B_{\epsilon'}^3 \cdot B_{\epsilon'} \cdot (d + \log(B_{\epsilon'}^4) + \log(1/\delta)) \cdot B_{\epsilon'}^4\right)$$
$$\leq O\left((d + \log(B_{\epsilon'}^4) + \log(1/\delta))B_{\epsilon'}^8\right).$$

Note that

$$B_{\epsilon'} \leq O\left(\log\left(\frac{\log(1/\epsilon)}{\epsilon}\right)\right) \leq O(\log(1/\epsilon)).$$

Thus combining the two bounds we get an upper bound for the number of queries:

$$q \leq O((d + \log\log(1/\epsilon) + \log(1/\delta)) \log^8(1/\epsilon))$$
$$\leq O((d + \log(1/\delta)) \log^9(1/\epsilon)).$$

Now we prove the correctness of the algorithm. First observe that by the definition of $m$ and the union bound over $O(B_{\epsilon'}^4)$ many events we know that success events of Lemmas 2 and 3 hold when lemmas are applied to functions of the form $\text{Maj}(h_1, \ldots, h_{i-1}), \text{Vote}_g(h_1, \ldots, h_{i-1})$, where $h_1, \ldots, h_t$ are functions constructed throughout the algorithm. Observe then that if for every $i \in I_{\epsilon'}, j \in [t]$ $\left(t = O\left(B_{\epsilon'}^3\right)\right)$ we define:

$$p_i^j := \mathbb{P}_{x \sim \mathcal{D}}[\text{Vote}_g(h_1, \ldots, h_j)(x) = i],$$

then $\{\{p_i^j\}_{i \in I_{\epsilon'}}\}_{j \in [t]}$ satisfies the rules of the process on $2\mathbb{Z} + 1$ with parameter $\epsilon'$ (Definition 4). Lemma 2 is responsible for the first property and Lemma 3 is responsible for the second property. Thus we can apply Lemma 4 to $\{\{p_i^j\}_{i \in I_{\epsilon'}}\}_{j \in [t]}$ to get that at the end of the process we have:

$$\sum_{i \in I_{\epsilon'}^+} p_i^t \leq 64 \cdot \epsilon' \cdot B_{\epsilon'}^3 \leq \epsilon,$$

where in the last inequality we used the assumption that $\epsilon < \frac{1}{32}$. To conclude observe that:

$$\sum_{i \in I_{\epsilon'}^+} p_i^t = \sum_{i \in I_{\epsilon'}^+} \mathbb{P}_{x \sim \mathcal{D}}[\text{Vote}_g(h_1, \ldots, h_t)(x) = i]$$
$$= R(\text{Maj}(h_1, \ldots, h_t)).$$

$\square$

We conclude by showing that Corollary 1 is a simple consequence of Theorem 1.

**Corollary 1.** *For every feature space $X$, for every $\epsilon \in \left(0, \frac{1}{32}\right)$, for every $d \in \mathbb{N}$, for every hypothesis class $\mathcal{H}$ on $X$ of VC-dimension $d$ there exists a learning algorithm* **L** *such that for every distribution $\mathcal{D}$, for every ground truth $h \in \mathcal{H}$, for every adversary* **A** *the following holds. When* **L** *interacts with* **A** *as described in Definition 3 then with probability $2/3$ at least one of the two properties holds:*

- **L**, *after $O(d \cdot polylog(1/\epsilon))$ rounds of interaction with* **A**, *returns a function $f$ such that $R_{\mathcal{D},h}(f) \leq \epsilon$,*

- *there exists $t \in [O(d \cdot polylog(1/\epsilon))]$ such that at the interaction round $t$ a function $f_t$ was presented to* **A** *and* $\mathbf{A}(f_t, EX_{\mathcal{D}}) \neq EQ_{\mathcal{D}}(f_t)$.

*Moreover, throughout the interaction only $O(polylog(1/\epsilon))$ different functions are presented to* **A**.

*Proof.* Let $X$ be a feature space, $\epsilon \in (0, \frac{1}{32})$, $\mathcal{H}$ be a hypothesis class of VC-dimension $d$. We will show that EQ-learner (Algorithm 1) satisfies the conditions of the corollary.

Assume that the EQ-learner is run with parameters $\epsilon, \delta = 1/3$, $\mathcal{H}$ and every call to $EQ_{\mathcal{D}}$ replaced by an interaction with **A**. This setup satisfies the requirements of the adversarial learning game (Definition 3). Now there are two possible scenarios. First scenario is that throughout the run of the algorithm, for all functions $f_t$ that **L** presents to **A** we have that $\mathbf{A}(f_t, EX_{\mathcal{D}}) = EQ_{\mathcal{D}}(f_t)$. Then Theorem 1 guarantees that with probability $2/3$ the first statement of the corollary is true. The other scenario is that there exists $f_t$ that **L** presented to **A** such that $\mathbf{A}(f_t, EX_{\mathcal{D}}) \neq EQ_{\mathcal{D}}(f_t)$. This implies the second statement of the corollary.

What is left is to observe that the EQ-learner queries the $EQ_{\mathcal{D}}$ only for $O(\text{polylog}(1/\epsilon))$ many different functions. This is true as the number of different functions sent to $EQ_{\mathcal{D}}$ is upper-bounded by $t \cdot (B_{\epsilon'} + 1) \leq O(\text{polylog}(1/\epsilon))$, where parameters $t$ and $B_{\epsilon'}$ are defined in the algorithm. $\square$

# B Experiments

To find adversarial examples for standard adversarial training and on-manifold adversarial training we used the attack from Madry et al. [2018]. When performing on-manifold adversarial training the search for perturbation is done in the latent space. The perturbations were bounded in the $\ell_\infty$ norm in both cases.

For training we use classifiers with three convolutional layers (4 × 4 kernels; stride 2; 16, 32, 64 channels), each followed by ReLU activations and batch normalization, and two fully connected layers. The networks are trained using ADAM Kingma and Ba [2014], with learning rate 0.01 (decayed by 0.95 per epoch), weight decay 0.0001 and batch size 100, for 20 epochs. For further details we refer the reader to Stutz et al. [2019]. We used the Google cluster for our experiments. It took 3 days to run.

Looking at the figures in Stutz et al. [2019] one might wonder why in some situations, in their language, the on-learned-manifold attack success rate is smaller than the test error. After all, the adversary always has the option of *not* perturbing the input. Hence, the test error is a lower bound on the attack success rate. One way this can happen is when the autoencoder maps an incorrectly classified input to a correctly classified one. More formally the autoencoder works as follows. An input $x$ is mapped to the latent space to produce $z = \text{enc}(x)$. Then the attacker tries to find a perturbation $\delta$ in the latent space such that $\mathcal{L}(\text{dec}(z + \delta), y)$, where $y$ is the label of $x$, is maximized. Let $f$ be the attacked classifier. Then it might happen that $f(x) \neq y$ but $f(\text{dec}(\text{enc}(x))) = y$. In this case it is a priori nontrivial to find a perturbation $\delta$ such that $f(\text{dec}(\text{enc}(x) + \delta)) \neq y$.