# OpenReview forum: "Exponential Separation between Two Learning Models and Adversarial Robustness"
_NeurIPS.cc/2021/Conference — NeurIPS 2021 Poster_

### Official Review · Reviewer_p28f · 2021-07-15

**Rating:** 6
**Confidence:** 3

**Summary:**

The paper studies the query complexity of learning a VC class under a certain query model called the Equivalence-Query (EQ) model by Angluin'98. In this model, given a distribution $\mathcal{D}$ and the ground truth $g$, the learner interacts with the data in the following way: (1) learner selects a function f and (2) learner receives a random counter-example distributed according to $\mathcal{D}$, that is, a random draw from $\mathcal{D}|_{f \ne g}$. Under this model, the authors prove an exponentially improved upper bound of $d \cdot \mathsf{polylog}(1/\epsilon)$ compared to the $d/\epsilon$ bound for PAC learning. The paper further draws connections between this model of learning and adversarial robustness. The main argument here relies on viewing the equivalence-query sample oracle as the adversary restricted to on-manifold attacks.

**Limitations And Societal Impact:**

Yes

**Main Review:**

Technical: The main separation result is technically novel and the analysis has some new ideas. The authors do a good job in putting their result in context with other known results in similar models. Beyond the main technical result the authors attempt to connect EQ-learning to on-manifold adversarial training and adversarial robustness. The viewpoint of equating an EQ oracle and an on-manifold adversarial training procedure is interesting. However, the connection to adversarial robustness is not convincing. This requires two assumptions (1) attacks are on the manifold (2) the attacks are distributed nicely (and not adversarially). Also, in the proposed model, the attacks need not even depend on the target image. I feel these assumptions are too strong for general real-world attacks. These attack points are more appropriately (correctly-distributed) errors due to generalization.

Writing: The writing of the paper needs improvement. There are a lot of high-level discussions but the technical aspects are lacking in detail. A more complete proof overview explaining the various different steps in the algorithm and their reasoning would help with readability. Removing tangential discussion in the main body would also help streamline the paper (especially in section 3.3 and 5).

Overall, I like the technical result on its own but am not convinced about the adversarial robustness viewpoint. Since a substantial part of the paper is focused on the latter, as is, I lean towards rejecting the paper.

Typos:
Line 69, Th $\rightarrow$ The
Equation above Line 596, index $i$ is overloaded

**Time Spent Reviewing:**

5

---

> ### Author Response · Authors · 2021-08-10
> **Connections to adversarial robustness**
>
> Thank you for the taking the time to review our paper. We address your questions about connections of our results to adversarial robustness.
>
> The standard approach to adversarial ML is to consider perturbation sets. The most commonly considered such perturbation sets stem from perturbations that are bounds in $\ell_p$ norms. There are perhaps thousands of papers following this approach. Whether those bounds capture real-world scenarios is up for debate but due to the sheer number of prior works it looks familiar to all of us. There is also the question whether this approach can lead to a satisfying solution in principle, given the many rounds of new defenses and attacks we seen over the past years.
>
> Rather than follow this well trodden path we wanted to escape the assumption of a fixed perturbation set and see if we could approach the problem in a principled way. This is akin to the approach taken in the paper by Goldwasser et al., 2020 where also the basic axiom of a bounded perturbation set was dropped. The model we introduce is in the spirit of a fixed adversary that we can interact with during training time. In fact we started our work on this paper by envisioning a system that gets better the more it is attacked.
>
> You are right by saying that our assumption that the adversarial examples come from a "uniform" distribution makes the analysis simple. Indeed, this is why we chose this model over other plausible such models. But we want to emphasize that all known adversarial attacks we know have the following property. On an input $x$ an attack looks for $\textbf{any}$ point in the neighbourhood of $x$ that fools the classifier. This means that there is nothing explicitly adversarial about the choice of the adversarial example apart from the fact that it is misclassified. An attack therefore induces a distribution on adversarial examples. To simplify things, we assumed that this distribution is uniform. But it is an assumption that is as canonical as any other (like the $\ell_p$ bound).
>
> Our approach achieves our initial goal of finding a principled solution in a simplified setting.
> What is missing is generalizing the framework to all distributions and where these distributions are chosen adversarially in each round. This might be the adversarial element that you noticed was missing in our model.
>
> We hope that this discussion convinced you that Section 3.3 and 5 are integral parts of the paper.

---

> > ### Comment · Reviewer_p28f · 2021-08-19
> > **Adversarial Robustness**
> >
> > Thanks for the detailed response. I agree with the motivation of developing new models of adversarial attacks however I do think that assuming that it is distributed according to $\mathcal{D}$ over the errors and independent of an anchor sample does not capture various settings in real-world such as targeted attacks where changing features of an input leads to a different outcome. Your motivation that usually algorithms choose any adversarial point from the neighborhood is correct however it is not according to the true data distribution and often not on the manifold.
> >
> > I would be willing to increase my score given that both the sections are made more precise and less tangential, and the overall writing is improved.

---

> > > ### Author Response · Authors · 2021-08-23
> > > **Adversarial robustness**
> > >
> > > Thank you for this comment. After seeing all the reviews we noted that our discussion of connections to adversarial robustness was not clear enough. In the final version of the manuscript we plan to focus more on the separation result and make the discussion about adversarial examples more concise and precise. We hope that this addresses your suggestions and we look forward to the improved score.

---

### Official Review · Reviewer_rx52 · 2021-07-17

**Rating:** 9
**Confidence:** 4

**Summary:**

This paper introduces a new abstract learning model called the EQ-learning, in which the oracle returns examples that the current classifier gets incorrect (rather than arbitrary examples drawn from the data distribution) and returns YES if no such examples exist (indicating the current classifier is correct). They then provide an algorithm which achieves a sample complexity (assuming finite VC dimension d)that is polylog in $1/\epsilon$ as opposed to $O(1/\epsilon)$. This is an exponential increase, and is a very interesting theoretical result on its own.

They then argue how their formalism plays a role in learning from on-manifold adversarial examples. In particular, each of these examples reveals a point that the classifier gets incorrect, and their result consequently suggests that learning from these results should improve the training procedure.

**Limitations And Societal Impact:**

Yes.

**Main Review:**

I think this is a very good paper. The abstract learning model is very elegant and simple while also being highly relevant to modern problems in practical machine learning. Their algorithm is interesting, and their result is terrific: showing an exponential gap is very exciting. This paper also spurs many interesting questions: how can their observations be leveraged into better examples for adversarial training? What is the relationship between adversarial training and generalization?

I strongly recommend this paper for acceptance.

**Time Spent Reviewing:**

2 hours

---

> ### Author Response · Authors · 2021-08-10
> **Thank you reply**
>
> Thank you for such a kind review of our paper.  We are happy that you agree that our work
> sparks interesting research directions.

---

### Official Review · Reviewer_Nq1S · 2021-07-17

**Rating:** 6
**Confidence:** 3

**Summary:**

The main result of the paper is showing a separation between the sample complexity of PAC model and the query complexity of an interactive model called Equivalence-Query-learning.
The interactive model requires exponentially fewer queries/samples (in the mistake parameter - epsilon).
Further, the implication of this result on the adversarial robustness is discussed.

**Limitations And Societal Impact:**

-

**Main Review:**

I have few questions:
1. I'm not sure why the "on-manifold" phenomenon in section 4 is explained by the main Theorem.
2. In section 5, you define some kind of different robust model. What is the implication on the standard robust PAC learning model?

Significance:
I'm not familiar with the  Equivalence-Query-learning model by Angluin, but the separation result between the aforementioned models seems interesting.
I understand the suggested attack in section 5, but I'm not sure what are the implications to the robust PAC learning model.

Clarity: Altogether, the overall method (for the EQlearner) is presented clearly.

**Time Spent Reviewing:**

3

---

> ### Author Response · Authors · 2021-08-10
> **Clarifications**
>
> Thank you for the taking the time to review our paper. Next we address your questions.
>
> 1. If we understood your question correctly you were asking why the main theorem explains the experimental finding from Section 4. We'll try to answer this question. In Section 4 we present an experiment which shows that using on-manifold adversarial learning improves generalization in comparison to standard training. Our theorem says (informally) that if the counterexamples given to the learning algorithm are distributed according to
>  $D_{f  \neq h}$ then that improves generalization in comparison to standard training. Thus we see that both statements have the same flavor. They are different in two aspects: the learning algorithms are different (one is just a neural network trained with adversarial examples while the other is our majority vote-like procedure) and the distribution on adversarial examples in the experiment might be different than $D_{f \neq h}$. We make two assumptions: the distribution on adversarial examples in the experiment is close to $D_{f \neq h}$ and the generalization behavior of the two algorithms is similar. Under these two assumptions the main theorem is a theoretical justification of the experimental results.
>
> 2. Implications of our results for the standard robust PAC learning model is an interesting research direction. Right now we don't have a satisfying answer. The two models are different and it is not obvious how to compare them. For instance, in our model we bound the learning power of the adversary while in the standard robust PAC learning it is unbounded.

---

### Decision · Program_Chairs · 2021-09-28

**Decision:**

Accept (Poster)

**Comment:**

The paper shows an exponential separation of sample complexity between a "version" of EQ learning and PAC learning. The "version" is indicated as such because the original introduction of EQ learning was in the context of exact learning (with MQs and EQs) and did not require any distributional assumptions on the data. Here the assumption is that the counterexamples come from the original distribution conditioned on the region of disagreement. While this should mean that the sample complexity gap (wrt PAC learning) should not be completely surprising, the paper does provide a neat argument and requires significant insight.

The connection to adversarial examples seems tenuous at best to me and to several of the reviewers (as well as in an earlier review process as per the authors). I would strongly suggest that the authors remove all unnecessary parts there, and focus more clearly on demonstrable connections to adversarial examples, or simply use the space to expand on the (interesting) theory results they already have.

**Consistency Experiment:**

NeurIPS has a long history of experimentation. In 2014, NeurIPS ran an experiment in which 10% of submissions were reviewed by two independent committees to quantify the randomness in the review process. This year, we repeated a variant of this experiment to see how the quality of the review process has changed over time.  This paper was part of the experiment and was therefore assigned to two committees (consisting of reviewers, an Area Chair, and a Senior Area Chair) that reached independent decisions.  If both committees made the same recommendation, this recommendation was followed. If a single committee recommended acceptance, the paper was accepted (with the exception of a few cases in which the other committee identified what we considered a fatal flaw, e.g., an error in a key result).

This copy’s committee reached the following decision: **Accept (Poster)**

The other committee assigned to the paper recommended **Reject**.  You can find the other set of reviews, along with any follow up discussion with the authors here:
https://openreview.net/forum?id=uRwcGRwQK1L